# Using Random Effect Models to Produce Robust Estimates of Death Rates in COVID-19 Data

**DOI:** 10.3390/ijerph192214960

**Published:** 2022-11-14

**Authors:** Amani Almohaimeed, Jochen Einbeck, Najla Qarmalah, Hanan Alkhidhr

**Affiliations:** 1Department of Statistics, College of Science, Qassim University, Buraydah 51482, Saudi Arabia; 2Department of Mathematical Sciences, Durham University, Durham DH1 3LE, UK; 3Department of Mathematical Sciences, Princess Nourah bint Abdulrahman University, Riyadh 11671, Saudi Arabia; 4Department of Mathematics, College of Science, Qassim University, Buraydah 51482, Saudi Arabia

**Keywords:** count data, Poisson model, death rates, case rates, robustness, random effects, mixture model, EM algorithm, shrinkage, MAP rule, COVID-19

## Abstract

Tracking the progress of an infectious disease is critical during a pandemic. However, the incubation period, diagnosis, and treatment most often cause uncertainties in the reporting of both cases and deaths, leading in turn to unreliable death rates. Moreover, even if the reported counts were accurate, the “crude” estimates of death rates which simply divide country-wise reported deaths by case numbers may still be poor or even non-computable in the presence of small (or zero) counts. We present a novel methodological contribution which describes the problem of analyzing COVID-19 data by two nested Poisson models: (i) an “upper model” for the cases infected by COVID-19 with an offset of population size, and (ii) a “lower” model for deaths of COVID-19 with the cases infected by COVID-19 as an offset, each equipped with their own random effect. This approach generates robustness in both the numerator as well as the denominator of the estimated death rates to the presence of small or zero counts, by “borrowing” information from other countries in the overall dataset, and guarantees positivity of both the numerator and denominator. The estimation will be carried out through non-parametric maximum likelihood which approximates the random effect distribution through a discrete mixture. An added advantage of this approach is that it allows for the detection of latent subpopulations or subgroups of countries sharing similar behavior in terms of their death rates.

## 1. Introduction

Since the beginning of the reporting of daily case and death numbers for the COVID-19 pandemic, the communication of death *rates* has been a non-trivial task. While there is no uniqueness in defining such a concept, we consider, in this work, such rates to mean a measure of daily deaths in relation to daily case numbers. This measure is, in the view of the authors, the most direct and meaningful way of defining a death rate: it is informative for the proportion of people who die of COVID-19 if infected, or, from the viewpoint of an individual, could be considered as a measure of the chance of dying of a COVID-19 infection, once infected.

The difficulties with this approach are obvious. Firstly, given a certain number of deaths on some calendar date of interest, it is not so clear in relation to cases on which date they should be set [1]. Secondly, whatever reference date for the cases is taken, it is possible that either the cases (denominator) or the deaths (numerator) take the value 0. In the former case, the raw death rates would be undefined, so that one could not even produce an estimate. But even the latter case, where the raw death rate would be zero, could be deemed practically useless, since there is solid empirical evidence that any lower bound on the case fatality rate is larger than zero [1,2]. Such problems are more likely to occur for small countries or entities than for large ones, but can also occur for larger countries, due to reporting issues of all kind (data transfer or entry issues, weekends, holidays, etc.).

Such issues are presumably the reason that most databases do not make an attempt to report “deaths out of cases” rates. For instance, on worldometer [3], one finds, for the whole world or per country/entity, cumulative numbers of “Deaths/1M pop”, and (at least for some of the countries) graphs of “Cumulative total deaths over cumulative number of closed cases”. While such cumulative information is helpful—for instance, one can infer from this that, at the moment of writing, the cumulative death rate in Saudi Arabia stands at 1.17%, as opposed to, say, 0.50% in France and 5.91% in Mexico—they do give very little insight into the current state of the pandemic, since it takes a long time for the cumulative death rates to adjust once the daily rates start shifting. Hence, in order to understand better the *current* state of the pandemic, rather than its cumulative picture since its beginning, a mechanism to produce daily estimates of death rates is required.

The interpretation of such rates needs clarification. Our objective is not to deal with phenomena such as non-reporting, under-reporting, possibly over-reporting of deaths or cases, or with adjusting for calendar effects such as weekends or holidays. Such matters have been discussed elsewhere [1,4,5]. In this article, no attempt is made to adjust for the substance of such effects. Instead, we postulate that “reported death numbers” and “reported case numbers” are quantities of interest in their own right. This is certainly the case: firstly, reported numbers are all what is available, and secondly, the comparison of, say, the reported case numbers in some country on a Tuesday to the case numbers on the Tuesday a week ago, is clearly a meaningful operation. In fact, the media commonly use comparisons of this type to make statements on the development of the pandemic.

But if that is so, then the ratio of these death and case numbers will also be of general interest. However, as previously explained, the *raw* ratios of reported death and case numbers, using whatever time lag between those, will often not yield a useful result. Hence, we think of this raw rate just as a (poor) estimate of some underlying construct, to which we refer as the “latent death rate”, which we would like to recover. An intuitive approach of doing so would leave the raw rate largely unchanged where sufficient information is available, but will “borrow” information from other countries where this information is lacking. That is, one can aim to leverage the heterogeneity between countries in order to obtain more reliable estimates for those where the uncertainty in the estimated raw rates is large. The remainder of the paper is dedicated to the development of a methodology which does exactly this.

While this paper takes a “statistical modeling” point view, the type of modeling is different to the majority of other contributions to the literature, which focus on modeling the transmission mechanisms or the temporal dynamics of COVID-19 infections. A full review of such approaches is rather elusive, with thousands of published papers existing. We mention here representatively [6] which models COVID-19 transmissions through a mathematical model based on fractional derivatives. Our work is different to this body of literature as we do not make any attempt at modeling the dynamics of COVID-19; our approach is entirely based on modeling cross-sectional heterogeneities at given time points.

The rest of the paper unfolds as follows. In Section 2, we explain the statistical methodology used to produce an estimator of the latent death rate. Section 3 deals with technical and subject-matter details such as the specification of the lag size between cases and deaths. Section 4 applies the method on recent COVID-19 data for a variety of scenarios. Section 5 concludes the exposition. Some additional output tables as well as the complete code, which enables applied users to reproduce all analyses presented in this paper as well as to carry out their own analyses, are given in the online Appendix A. The code is implemented in the statistical programming language R [7] and is provided in form of a markdown notebook.

## 2. Methodology

Our methodological approach is rooted in the small area estimation literature, and can be traced, at least in spirit, to work by [8] and earlier related references therein. However, we extend their approach towards a two-stage methodology (to deal with both cases and deaths) and use a different fitting technique.

Let us firstly revisit the one-stage problem in which we are only talking of cases (not yet of deaths). This will serve as our “upper model”. Here, given a set of counts $y_{i}$ from regions (countries) i=1,…,m, with population sizes $n_{i}$, i=1,…,m, we can think of the $y_{i}$ as realizations of random variables
(1)$$Y_{i}∼Pois\left(\mu_{i}\right),$$
where the Poisson mean parameters $\mu_{i}$ are modeled by
(2)$$\log\mu_{i}=\log n_{i}+z_{i},$$
with the random effects $z_{i}∼Z$ following some distribution *Z*. Equation (Equation 2) corresponds to the natural way of modeling the Poisson mean parameter using a log-link under the presence of an offset [9], but with the particular feature that the parametric linear predictor, which would typically be described by an expression of type $x_{i}^{T}\beta$, is here entirely replaced by the random effect, $z_{i}$. The ratios $r_{i}=\mu_{i}/n_{i}$ can be considered as latent case rates. These will be estimated, in accordance with the model, by $\exp({\tilde{z}}_{i})$, where the ${\tilde{z}}_{i}$ will be obtained as posterior random effects from the fitted model.

The distribution of *Z* will account for the heterogeneity of the population of countries in terms of the expected (reported) COVID-19 counts on a single, given day. It is noted that, of course, the $\mu_{i}$ will also be depending on time (day), *t*, so that one could write $\mu_{it}$. However, we will be considering always only one time point on its own, so that we can suppress the notation *t*.

From Equation (Equation 2), one observes that the expected country-wise counts, $\mu_{i}$, depend on the population sizes, but also on other latent factors: different countries have different demographics, different health systems, and reacted in fundamentally different ways to the rise of the pandemic; with some countries taking stringent measures while some taking a more liberal approach. The point to make is that it would be wrong to consider the Poisson counts from each country as stemming from the same distribution, even after adjusting for different population sizes $n_{i}$. There is substantial heterogeneity which is taken into account by the random effect *Z*.

There is some flexibility in how precisely this is accomplished. In [8], the distribution of *Z* is assumed Gaussian, and the posterior random effects ${\tilde{z}}_{i}$ are obtained in a fully Bayesian manner. In our context, there is little reason to think that the $z_{i}$ are behaving “normally”. Due to the large heterogeneity between countries, and different but sometimes related circumstances therein (e.g., geographically neighboring, or similar COVID-19 measures were taken), it would rather feel appropriate that the distribution of *Z* is clustered; i.e., one could think of it as a mixture distribution. We therefore follow the approach laid out by [10] in which the distribution of *Z* is not parametrically specified, but nonparametrically estimated from the data, as a mixture distribution with *K* mass points $z_{k}$, k=1,…,K, and associated mixture probabilities $p_{1},\ldots ,p_{K}$. The value *K* represents the number of latent clusters or subpopulations as postulated by the data analyst; it can be informed by model selection criteria as discussed in the next section. The mixture parameters are estimated through the Expectation-Maximization (EM) algorithm, in a procedure known as “Nonparametric Maximum Likelihood” (NPML), yielding estimates ${\hat{z}}_{1},\ldots ,{\hat{z}}_{K}$, ${\hat{p}}_{1},\ldots ,{\hat{p}}_{K}$. From the fitted model, posterior random effects ${\tilde{z}}_{i}$ can be obtained through Empirical Bayes Predictions [11], as ${\tilde{z}}_{i}=\sum_{k=1}^{K}w_{ik}{\hat{z}}_{k}$, where the $w_{ik}$ are the “responsibilities” (i.e., posterior probabilities that observation *i* belongs to mass point ${\hat{z}}_{k}$) extracted from the EM algorithm after convergence. Completing this procedure leads to “shrunk” rates
$${\tilde{r}}_{i}=\exp\left({\tilde{z}}_{i}\right)$$
which can be considered as robust counterparts of $y_{i}/n_{i}$. One can think of these ${\tilde{r}}_{i}$ as achieving a compromise between assuming, on the one hand, a constant rate $r_{i}\equiv r=\sum_{i=1}^{m}n_{i}y_{i}/\sum_{i=1}^{m}n_{i}$ across the whole world, and, on the other hand, allowing for the country-wise data to speak completely by themselves, as $y_{i}/n_{i}$. The less information is available for country *i*, in terms of the population size $n_{i}$, the more the raw count for this country will be shrunk towards *r*. The larger $n_{i}$, the closer ${\tilde{r}}_{i}$ will be to $y_{i}/n_{i}$. A similar approach has been followed by [12] to model suicide rates over several Health Boards of the Republic of Ireland.

However, unlike this and related papers [13,14], we are going now one step further. Recall that eventually we are interested in estimating death rates out of cases. Hence, we build a “lower model” where observed deaths $d_{i}$ in country *i*, i=1,…,m can be considered as realizations of random variables
(3)$$D_{i}∼Pois\left(\lambda_{i}\right),$$
where the Poisson mean $\lambda_{i}$ is now modeled as
(4)$$\log\left(\lambda_{i}\right)=\log\left({\tilde{y}}_{i}\right)+c_{i}.$$

The random effects $c_{i}∼C$ follow an unspecified distribution *C*, and $\log({\tilde{y}}_{i})$ acts as an offset with ${\tilde{y}}_{i}$ being estimated from the upper model fit as
(5)$${\tilde{y}}_{i}=n_{i}\exp\left({\tilde{z}}_{i}\right)=n_{i}{\tilde{r}}_{i}.$$

In order to fit this model, we use mass points $c_{1},\ldots ,c_{L}$ with masses $q_{1},\ldots ,q_{L}$, where *L* reflects the number of subpopulations assumed on this level, to approximate the distribution of *C*. As before, this yields posterior random effects ${\tilde{c}}_{i}=\sum_{\ell =1}^{L}v_{i\ell }{\hat{c}}_{\ell }$, with corresponding responsibilities $v_{i\ell }$. Then, one can estimate the latent death rates $s_{i}\equiv \lambda_{i}/{\tilde{y}}_{i}=\exp\left(c_{i}\right)$ by shrunk rates
$${\tilde{s}}_{i}=\exp\left({\tilde{c}}_{i}\right).$$

The use of ${\tilde{y}}_{i}$ in the linear predictor (Equation 4), and hence in the denominator of the ratio $s_{i}$, has two benefits: firstly, we deduce from (Equation 5) that ${\tilde{y}}_{i}$ is always positive provided $n_{i}>0$, ensuring the existence of this ratio. Secondly, it reduces unwanted volatility of this ratio due to high fluctuations in the $y_{i}$ in the presence of small counts.

## 3. Technical Details

### 3.1. Choice of Lag

The epidemiological time delay dynamics of COVID-19 infections must be studied and followed in order to detect changes in the virus’s phenotype, the effectiveness of treatment, the affected demographics, and the ability of the healthcare system to handle large numbers of patients [15]. One of the most crucial quantitative measures is the case fatality rate. However, case rates and death rates are out of sync, since there is some lag (delay) between (the reporting of) a case and (the reporting of) a death. Hence, establishing the “death out of cases” rate requires a precise estimate of the time lag from infections to deaths [16].

We have carried out a literature search on this question, involving 13 sources involving data from 7 countries. Table 1 summarizes the results from this literature search. Based on this, we believe that there is solid evidence in the literature that a reasonable estimate of the time from onset to death is in the region of 14 days (unweighted mean of the values in the 4th column: 14.0; median: 15.2). There are some studies which appear to deviate strongly from this, but these are studies with small sample sizes, and/or with data from very early in the pandemic. In any case, a lag size which is a multiple of 7 appears to be a sensible choice since “reporting effects” for both cases and deaths tend to follow a weekly pattern.

In some situations, the study of a death rate involving a *zero* lag may still be useful, as it provides the most current snapshot of the pandemics available, even though the temporal link between cases and deaths is clearly broken under such a setting.

In the applied parts of the paper which follow, we concentrate on zero lags and 14-day lags.

### 3.2. Choice of Number of Mass Points

The procedure requires the selection of both *K* and *L*. Based on extensive studies in which we minimized the Bayesian Information Criterion (BIC) separately over both model fits for a variety of dates and lags, we recommend the general settings K=30 and L=4. This reflects a high heterogeneity between countries in the case rates, as opposed to a lesser heterogeneity in the death rates, given the fitted case rates.

### 3.3. Clustering and MAP Rule

In order to deal with heterogeneity of countries in terms of their death rate, the provided methodology uses a small number of mass points, $c_{1},\ldots ,c_{L}$, with L=4 being suggested in the previous subsection. These mass points can be interpreted as capturing latent subgroups or clusters of countries in terms of the risk of dying of a COVID-19 infection. In Section 2, we already mentioned the responsibilities
(6)$$v_{i\ell }=P\left(class\;\ell |country\;i\right),$$
which can interpreted as the posterior probability that country *i* belongs to class *ℓ*, ℓ=1,…,L. Here, the word posterior can be interpreted from a Bayesian perspective, noting that the responsibilities $v_{i\ell }$ are computed using Bayes’ theorem [29]. Using the available data, and conditional on the fitted model, they quantify the weight of evidence that country *i* belongs to any of the latent subgroups ℓ=1,…,L. Hence, we can also identify that subgroup for which this weight of evidence is maximal.

This concept enables the possibility of clustering countries in terms of their COVID-19 mortality. (Of course, the same could also be done for the case rates; however, the heterogeneity between countries is here much larger, and clustering a country into one of, say, K=30 groups is unlikely to be practically useful. Hence, we will not consider this possibility further.) The clustering step is executed easily as an add-on step to the EM algorithm, by applying the MAP (maximum a posteriori) rule onto each row of the matrix $V=(v_{i\ell })$:$$\hat{\ell }\left(i\right)={argmax}_{\ell }v_{il}$$

That is, each country, *i*, is assigned to the mass point (cluster center), $\hat{\ell }\left(i\right)$, to which it belongs with maximum posterior probability, given the fitted model. In simpler words, it is the most likely class for country *i*, according to our model. The matrix *V* is available as a by-product from the last E-step of the EM algorithm, and the ensuing clustering is fully data-driven, i.e., it does not require subjective judgements on the cluster allocation. One could, however, argue that one should additionally consider and report the uncertainty in the cluster allocation. For country *i*, this uncertainty is small when one of the $v_{i\ell }$’s is close to 1, and large if the two largest $v_{i\ell }$’s are quite similar. Reference [12] suggests a graphical format of representing this uncertainty. Further detail on the MAP methodology in the context of mixture models is given in [29].

## 4. Results

The data used in this section are publicly available from [30], using the code as provided in the online Appendix A.

### 4.1. Robust Rates for All Countries

First, we consider the results of a single execution of the methodology outlined in Section 2. We choose for this the 21 June 2022, which is chosen for no particular reason except that it is relatively recent, and that it is a Tuesday, hence largely avoiding weekend effects (even though there may still be overspill effects from weekend under-reporting, but one can argue that any those effects are likely to affect both cases and deaths, so are unlikely to have a major impact on death *rates*). We apply the methodology with K=30 and L=4, requiring fitting two models. The two required EM algorithms take about 15 and 35 iterations, respectively, until convergence, requiring a total of 6–8 s computational time on an average laptop. We carry out the analysis for a lag of 0 and a lag of 14, and report the results for the zero lag in Table 2, and for the 2-week lag in Table 3. The methodology delivers, at one go, robust death and case rates for all 221 countries in the data base, but for reasons of space, we display here, alphabetically, only the first 20 each (from Afghanistan to Belize). The full tables are provided in the Appendix A.

From the tables, we draw the immediate observation that fitted case and death rates are always available, even if case or death numbers are 0 and even if the raw death rate is NaN. Furthermore, we observe that, while there is considerable variability in the case rates (both raw and fitted), the variability in the fitted death rates is much lower, with most fitted rates hovering around the grand rate of about 0.0020, but some exceptions (such as, considerably smaller death rates in Bahrain). For most countries, there are no massive differences between the unlagged and the lagged version, but here again there are some exceptions as, for instance, in Bangladesh where the case rate was considerably lower 14 days before, hence yielding a higher death rate under the lagged estimation.

### 4.2. Finding Clusters of Countries

Reconsidering the 14-day lag model fitted in the previous subsection, Table 4 displays the first 20 rows of the matrix $V=(v_{i\ell })$. The last column of this table presents additionally the MAP classifications according to the methodology outlined in Section 3.3. The table also presents, in the bottom two rows, the fitted mass points ${\hat{c}}_{\ell }$ and the corresponding masses ${\hat{q}}_{\ell }=\frac{1}{m}\sum_{i=1}^{m}v_{i\ell }$. These masses can be interpreted as the overall probability (not conditioning on the observed death count for country *i*) that any country is associated to cluster *l*. We observe that the mass points increase with ℓ,ℓ=1,…,4, so that, in view of model (Equation 4), clusters of increasing index *ℓ* correspond to higher death rates. From this table, we see that, for countries where little information was available from the data, the posterior probabilities are liberally spread over the four mass points and coincide approximately with the overall mixture probabilities ${\hat{q}}_{\ell }$, hence providing a “default” MAP allocation to cluster 2 for such countries. In the excerpt of countries that we have in front of us in Table 4, most countries are allocated to this default cluster, with only Afghanistan and Bangladesh allocated to cluster 4 (the cluster associated with highest death rates), and Belgium and Bahrain to cluster 1 (smallest death rates). Overall, 26 countries were allocated to cluster 1, 158 countries to cluster 2, 9 countries to cluster 3, and 28 countries to cluster 4. The full table of posterior probabilities and classifications is given in the Appendix A. We omit the results for the zero day lag analysis which would proceed along the same lines, but leave it with the comment that we found the results arising from the 14-day lag slightly more meaningful in terms of the ensuing classification.

### 4.3. Case Study 1: San Marino Data in 2021

In order to illustrate our methodology from a different angle, let us additionally look at an exemplary, very small country, which has been plagued by high COVID-19 case numbers from the very first weeks of the pandemic on: the Republic of San Marino. Figure 1 (top row) shows the daily case and death number in San Marino over the year 2021. Taking raw daily rates at face value (without taking lags), out of the 365 possible daily death rates, 151 are not computable at all (the R programme returns NaN); 180 return the value 0, and only 34 return a meaningful value between 0 and 1. When taking 14-day lags for the case numbers, then 146 rates are NaN, 185 are 0, and again only 34 produce an actual ratio. Of course, a death ratio of 0 does not need to be wrong per se. Still, it is likely to be an artifact of small counts, and it is also implausible from what we know about the pandemic. Similarly, also the 34 non-zero rates which did get produced are likely to suffer from high volatility, and hence little reliability, due to small counts both in the numerator and in the denominator.

After applying the methodology introduced in the previous section (here using a 0-day lag), one finds that robust death rates (out of cases) are computable for all 365 days of the year 2021. The fitted case rates (out of population; left) and death rates (out of cases; right) are provided in the second row of Figure 1. We observe from this that little shrinking appears to take place for the case rates: the top left panel in Figure 1 looks very similar to the middle left panel, with a pattern of moderately high but volatile case rates in the initial 100 days of 2021, followed by 200 days of very low case rates, and then strongly increasing case rates towards the end of 2021, in line with the global Omicron outbreak. It is notable that the robust death rates (out of cases), in the middle right panel of Figure 1, are largely unaffected by what happens to the case *rates*. Instead, the death rates follow their own pattern, with a mildly oscillating but overall decreasing wave-like pattern. It is again highlighted that these patterns in the death rates cannot be made visible using raw rates as they are simply not meaningfully computable for the majority of days. While inevitably still variable due to daily reporting differences and the lack of temporal smoothing, it is clear that the produced robust death rates appear meaningful, interpretable, and consistent with what one would have expected to see from our knowledge of the general flow of the pandemic.

The bottom row of Figure 1 gives plots of fitted versus raw case rates (left) and death rates (right; plotted only those days for which raw death rates are not NaN). The plots confirm that the methodology only mildly adjusts the case rates, but substantially modifies the raw death rates, while still correctly displaying the general trend (as evidenced by the red line of equality of rates, passing straight through the middle of the black dots).

### 4.4. Case Study 2: Saudi Arabia Data in 2021

In order to compare the results obtained in the San Marino case study with a scenario where we have a large population size in the country of interest, we additionally give results for an equivalent analysis carried out with the Kingdom of Saudi Arabia (population about 36 M). We firstly note that, despite the large population size, there seems to be still some justification to apply this methodology: for a 0-day lag, we have now 46 out of 365 non-computable (NaN) rates, and 319 computable non-zero rates. For the 14-day lag, these numbers would stand at 22 non-computable rates, 24 zero rates, and 319 computable non-zero rates.

These results are provided, in the same arrangement as before, and again using a 0-day lag, in Figure 2. To some extent, the results are similar to the results given above. The ability of the method to produce meaningful death rates over a time range spanning from around day 205 to day 270, where we have very many observed zero death counts, is notable. A moderate shrinkage of estimated death rates can also be observed from the bottom right panel.

## 5. Discussion

In this work, we were interested in estimating daily death-out-of-cases COVID-19 rates. In Epidemiolgy, a commonly used term in this context is the case fatality rate (CFR), which is the proportion of people who die from a specified disease among all individuals diagnosed with it. In this sense, one can consider the proposed methodology as a device of computing *daily* CFRs. However, since we have argued in the introduction that the construct being estimated through our approach (latent death rates) has a somewhat specific interpretation, we intentionally downplayed the term CFR in this exposition.

The estimation of event rates from count data is a problem with a long history in both the methodological and applied statistical literature. The problem can be related directly to the definition of the Poisson distribution as such: if events happen at a certain constant event rate *e* over a fixed time interval Δt, then the distribution of the number of events in the time interval Δt is known to be Pois(eΔt) [31]. In the context of this manuscript, the time interval Δt is always “1 day”, so that we can identify $\lambda =e\Delta_{t}$, without a time-length component depending on *t*. However, this does not take away the need to consider carefully the required model for λ, which may need to depend on covariates [9,12], random effects [12,29] (to address heterogeneity), and an offset [9,12] (to set events in relation to the size of a source population). Our approach comprises the latter of these two features. However, in the situation of estimating COVID-19 fatality rates “out of cases”, as considered in this manuscript, the offset would be the logarithm of a quantity which could be either zero or very small, leading to offsets which are either undefined or highly volatile.

Therefore, as a new methodological contribution for the estimation of death rates in the presence of (potentially) small case counts, we introduce a two-stage modeling process in which the case rates (out of population) are modeled by an “upper” Poisson model themselves, leading to more robust and guaranteed positive fitted case numbers which can be used as offset in the “lower” model for the death rates. We have shown that this leads to meaningful and interpretable estimates of COVID-19 death rates which are always computable, even in the presence of zero case counts (harder problem) or death counts (easier problem). This is achieved by “borrowing” information from other units (countries), which can be considered as a shrinkage method which leaves the raw rates largely untouched if the case numbers are large, but shifts the estimated rates the closer to the grand mean, the less information is available, that is, the smaller the case number of an individual country is, given an appropriately defined lag.

While we have given the choice of lag time between (reported) onset and (reported) death time some attention in this paper, it is clear that the suggestion of a 14-day lag size is not the final word on the matter, and that this is a question which needs further involvement with the literature, in particular as this lag time may start shifting, for instance due to improved medical care. However, this question is not the main concern of this manuscript. We presented a methodology which can be applied to *any* lag size, including zero, and the question of deciding on the appropriate lag size will ultimately be in the hands of the data analyst, depending on subject matter considerations. There could also be some dependence of the lag size on the specific COVID-19 variant at play. Reference [15] investigated the time delay distribution from infection to a clinical outcome in the context of the growth in the B.1.1.7 variant in December 2020 and did not find evidence for such differences; however, they also acknowledged that further research should be conducted on the role of novel variants in this aspect. An interesting approach to the problem is to use the entire delay distribution for the analysis as in [2]. This approach arguably increases the robustness of the estimated fatality rates but still requires some assumptions and a more complex modeling approach.

The calculations in this manuscript have been carried out using a code implemented in the statistical programming language R [7]. The complete code to reproduce the analyses presented in this manuscript is provided in the Appendix A. The code given in there can be immediately used and adapted to compute COVID-19 death rates for other (such as current) COVID-19 data, and does of course allow to change tuning parameters such as the lag size. An online “R Shiny” app which allows to produce robust COVID-19 death rates at one click for any given country and date is under preparation.

As alluded to previously, we have not applied any temporal smoothing in this work. Under the presented setup, information is borrowed only cross-sectionally across countries, but not longitudinally over time. This explains why the fitted rates over time such as in Figure 1 still do not look really smooth. Asymmetric (i.e., left-sided) temporal smoothing can be realized through exponential smoothing [32]. Mixtures of kernel-based exponential smoothing methods have been developed in [33]. Extending the methodology proposed in the present manuscript to this framework is left for future research.

Finally, it is noted that there is no reason why the methodology could not be applied to obtain (say) weekly rather than daily death rates, or to compute robust fatality rates for other diseases, or even to compute robust failure rates in engineering-type problems. Such applications are also left for further investigation.

## Figures and Tables

**Figure 1 ijerph-19-14960-f001:**
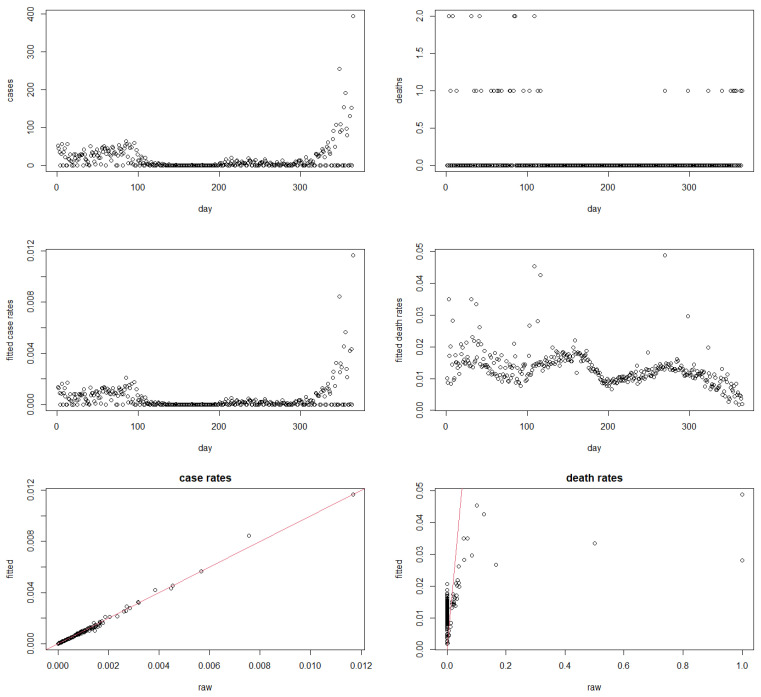
**Top**: daily COVID-19 cases (**left**) and deaths (**right**) in San Marino in 2021; **middle**: fitted daily COVID-19 rates (**left**) and death rates (**right**) in San Marino in 2021; **bottom**: fitted versus raw case rates (**left**) and death rates (**right**) in San Marino in 2021.

**Figure 2 ijerph-19-14960-f002:**
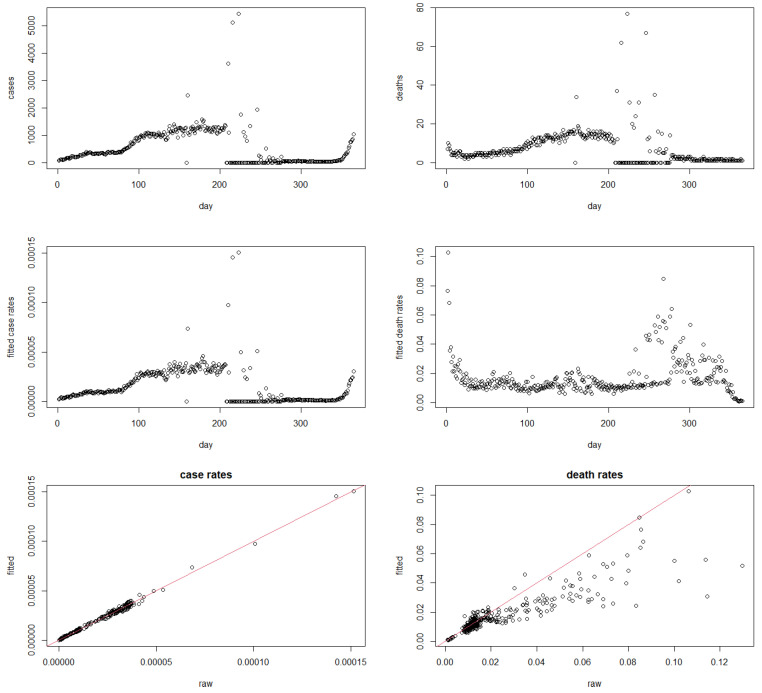
**Top**: daily COVID-19 cases (**left**) and deaths (**right**) in Saudi Arabia in 2021; **middle**: fitted daily COVID-19 rates (**left**) and death rates (**right**) in Saudi Arabia in 2021; **bottom**: fitted versus raw case rates (**left**) and death rates (**right**) in Saudi Arabia in 2021.

**Table 1 ijerph-19-14960-t001:** Lag from onset of symptoms to death for COVID-19 patients and the total numbers of deaths in each study.

Study ID	Region	Sample Size	Lag (Days)	Deaths (Count)
Zhou et al. (2020) [17]	China, Wuhan	191	18.5	53
Ruan et al. (2020) [18]	China, Wuhan	150	18.0	68
Jin et al. (2020) [19]	China, Wuhan	1056	13.0	37
Chen et al. (2020) [20]	China, Wuhan	50	13.0	50
Verity et al.(2020) [1]	China, mainland	3665	17.8	NA
Harrison et al. (2020) [21]	UK	7802	7	7802
Harrison et al. (2021) [22]	UK	1026	21	236
Faes et al. (2020) [23]	Belgium	14,618	9	1534
Hawryluk et al. (2020) [24]	Brazil	1,557,000	15.2	NA
Marschner (2021) [25]	Australia	6235	18.1	816
Lefrancq et al. (2021) [26]	France	198,846	19.0	33,269
Asirvatham et al. (2021) [27]	India	1761	4.0	1710
Mehta et al. (2021) [28]	India	346	9.0	76

**Table 2 ijerph-19-14960-t002:** Raw and fitted cases, case rates, deaths, and death rates, on Tuesday, 21 June 2022.

Location	Population	Cases	Fitted	Raw	Fitted	Deaths	Fitted	Raw	Fitted
			Cases	Case Rate	Case Rate		Deaths	Death Rate	Death Rate
Afghanistan	40,099,462	83	91.839	0.0000021	0.0000023	1	0.359	0.0120482	0.0039064
Albania	2,854,710	219	229.240	0.0000767	0.0000803	0	0.251	0.0000000	0.0010937
Algeria	44,177,969	8	2.766	0.0000002	0.0000001	0	0.005	0.0000000	0.0019397
Andorra	79,034	0	0.013	0.0000000	0.0000002	0	0.000	NaN	0.0019828
Angola	34,503,774	0	2.160	0.0000000	0.0000001	0	0.004	NaN	0.0019489
Anguilla	15,753	0	0.005	0.0000000	0.0000003	0	0.000	NaN	0.0019829
Antigua/Barbuda	93,220	0	0.014	0.0000000	0.0000002	0	0.000	NaN	0.0019828
Argentina	45,276,780	0	2.835	0.0000000	0.0000001	0	0.005	NaN	0.0019386
Armenia	2,790,974	0	0.183	0.0000000	0.0000001	0	0.000	NaN	0.0019801
Aruba	106,536	0	0.016	0.0000000	0.0000001	0	0.000	NaN	0.0019828
Australia	25,921,089	32,895	36,211.746	0.0012690	0.0013970	62	75.118	0.0018848	0.0020744
Austria	8,922,082	5233	5305.067	0.0005865	0.0005946	5	4.154	0.0009555	0.0007830
Azerbaijan	10,312,992	21	23.619	0.0000020	0.0000023	0	0.040	0.0000000	0.0016807
Bahamas	407,906	34	33.705	0.0000834	0.0000826	0	0.054	0.0000000	0.0015891
Bahrain	1,463,265	2078	2044.180	0.0014201	0.0013970	0	1.553	0.0000000	0.0007595
Bangladesh	169,356,251	874	973.659	0.0000052	0.0000057	2	1.425	0.0022883	0.0014636
Barbados	281,200	145	146.826	0.0005156	0.0005221	0	0.173	0.0000000	0.0011807
Belarus	9,578,168	0	0.600	0.0000000	0.0000001	0	0.001	NaN	0.0019734
Belgium	11,611,420	0	0.727	0.0000000	0.0000001	0	0.001	NaN	0.0019714
Belize	400,031	277	238.169	0.0006924	0.0005954	1	0.433	0.0036101	0.0018169

**Table 3 ijerph-19-14960-t003:** Raw and fitted case and death rates, on Tuesday, 21 June 2022, relative to cases 14 days earlier (i.e., Tuesday 7th of June). Population and death counts omitted since they are the same as in Table 2.

Location	Cases	Fitted Cases	Raw CASE Rate	Fitted Case Rate	Fitted Deaths	Raw Death Rate	Fitted Death Rate
Afghanistan	53	65.833	0.0000013	0.0000016	0.439	0.0120482	0.0066713
Albania	53	54.046	0.0000186	0.0000189	0.070	0.0000000	0.0012918
Algeria	4	3.914	0.0000001	0.0000001	0.007	0.0000000	0.0018068
Andorra	0	0.017	0.0000000	0.0000002	0.000	NaN	0.0018613
Angola	0	3.056	0.0000000	0.0000001	0.006	NaN	0.0018185
Anguilla	0	0.007	0.0000000	0.0000004	0.000	NaN	0.0018614
Antigua and Barbuda	0	0.019	0.0000000	0.0000002	0.000	NaN	0.0018612
Argentina	0	4.010	0.0000000	0.0000001	0.007	NaN	0.0018054
Armenia	0	0.273	0.0000000	0.0000001	0.001	NaN	0.0018576
Aruba	0	0.021	0.0000000	0.0000002	0.000	NaN	0.0018612
Australia	33,223	33,031.390	0.0012817	0.0012743	48.139	0.0018848	0.0014574
Austria	2183	2284.134	0.0002447	0.0002560	3.953	0.0009555	0.0017305
Azerbaijan	0	0.925	0.0000000	0.0000001	0.002	0.0000000	0.0018483
Bahamas	20	19.675	0.0000490	0.0000482	0.032	0.0000000	0.0016104
Bahrain	997	1054.167	0.0006814	0.0007204	0.340	0.0000000	0.0003224
Bangladesh	54	60.609	0.0000003	0.0000004	0.722	0.0022883	0.0119116
Barbados	104	93.738	0.0003698	0.0003334	0.099	0.0000000	0.0010537
Belarus	0	0.861	0.0000000	0.0000001	0.002	NaN	0.0018492
Belgium	5944	6189.447	0.0005119	0.0005330	1.182	NaN	0.0001910
Belize	224	212.846	0.0005600	0.0005321	0.638	0.0036101	0.0029986

**Table 4 ijerph-19-14960-t004:** Matrix *V* for data from 21 June 2022 using the 14-day lag, along with MAP classifications (right column) and the associated masses and mass points (bottom two rows).

*ℓ*	1	2	3	4	$\hat{\ell }$
Afghanistan	0.020	0.174	0.230	0.577	4
Albania	0.313	0.369	0.172	0.145	2
Algeria	0.259	0.324	0.174	0.243	2
Andorra	0.254	0.320	0.174	0.252	2
Angola	0.258	0.323	0.174	0.245	2
Anguilla	0.254	0.320	0.174	0.252	2
Antigua and Barbuda	0.254	0.320	0.174	0.252	2
Argentina	0.259	0.324	0.174	0.243	2
Armenia	0.254	0.320	0.174	0.251	2
Aruba	0.254	0.320	0.174	0.252	2
Australia	0.000	1.000	0.000	0.000	2
Austria	0.000	0.839	0.161	0.000	2
Azerbaijan	0.255	0.321	0.174	0.250	2
Bahamas	0.277	0.340	0.175	0.208	2
Bahrain	0.746	0.247	0.007	0.000	1
Bangladesh	0.000	0.025	0.099	0.875	4
Barbados	0.351	0.392	0.164	0.093	2
Belarus	0.255	0.321	0.174	0.250	2
Belgium	1.000	0.000	0.000	0.000	1
Belize	0.054	0.397	0.348	0.201	2
${\hat{c}}_{\ell }$	−8.564	−6.531	−5.458	−4.251	
${\hat{q}}_{\ell }$	0.254	0.320	0.174	0.252	

## Data Availability

We make use of publicly available data which can be downloaded from https://covid.ourworldindata.org/data/owid-covid-data.csv, accessed on 1 September 2022. The markdown file in the Appendix A illustrates how our code automatically downloads the data from this source.

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
