# Peer review of "Using Random Effect Models to Produce Robust Estimates of Death Rates in COVID-19 Data"

_ijerph, 2022, doi:10.3390/ijerph192214960_

Round 1

Reviewer 1 Report

Referee report on the manuscript (ID:ijerph-1928476)“Using random effect models to produce robust estimates of death rates in Covid-19data”

The authors of this manuscript proposed two Poisson models with random
effect to model the Covid-19 case numbers and Covid-19 death numbers,
respectively. The proposed models are shown to generate robust estimates
of Covid-19 cases and Covid-19 deaths, which can then be used to estimate
the Covid-19 death rates. The research problem is clearly defined and the
models are properly constructed with well-designed inference methods. The numerical examples also illustrate the effectiveness of the proposed methods.

The paper is well written. I only have some minor comments:

1. When motivating the research problem, the authors need to explain the
issues more clearly. For example, the authors made some comments on
page 8 in lines 200-201. Those comments should appear much earlier in
this manuscript, for example on page 1 in the second paragraph of Sec-
tion 1. Otherwise, readers might doubt the necessity of concerning zero
death rates. It would be better if the authors could escalate the issues
in empirical death rates estimation using the Covid-19 data from the
medical science point of view, eg why it isn’t meaningful/appropriate
to have zero death rates for Covid-19.

2. At the start of Section 3.2 the authors discussed K and L directly
without any explanations. Actually, I checked Section 2 very carefully
to find what these two constants are. K appeared once in the middle of
page 3 and L appeared after (5), both without any formal definition or
explanation. I suggest the authors to introduce them formally before
use them in the model constructions.

Overall, I quite enjoy reading this manuscript and I believe the models
are of practical interests to many people.

Author Response

  1. We would like to thank the referee for this important comment. Some new pieces of text, including two sentences at the position which the referee suggests, have been added to the introduction.
  2. We have given the definitions of K and L more explicitly where they occur first, and added additional explanations. 

Reviewer 2 Report

SummaryAuthors offered a new methodology for the estimation of death rates in the presence of small case counts with introducing a two-stage modeling process in which the case rates (out of population) are modeled by an "upper" Poisson model themselves, leading to more robust and guaranteed positive fitted case numbers which can be used as offset in the "lower" model for the death rates.

Major comments: 

1. While authors explained the impact of their model using a case study for a very small country with small population size, it would make more sense to also evaluate their methodology and its usefulness for larger population size. Therefore, I recommend to add another case study for a larger population size.

2. To add robustness to their model, authors need to comment on the impact of different phases of pandemic (the impact of different variants) on their model and specifically on their choice of lag.

3. The manuscript would benefit from more details regarding the unbiased "clustering of countries" and the MAP rule. With the current explanation, it seems the concept of clustering is more subjective.

Author Response

  1. Thank you for this suggestion. We have added a new Section 4.4 with a second case study, for Saudi Arabia.
  2. This is an important comment which is however not easy to address for us. We have added some discussion and references on this matter to Section 5.
  3. We have expanded the presentation in Section 3.3.

Reviewer 3 Report

Reviewer Comment

Title
: Using random effect models to produce robust estimates of death rates in Covid-19 data
Autho
rs: Amani Almohaimeed, Jochen Einbeck, Najla Qarmalah and Hanan Alkhidhr

Journal: International Journal of Environmental Research and Public Health

Random effect models are proposed to investigate death rates in Covid-19 data. The proposed model has been fully analyzed. Some examples are given to illustrate the effectiveness of the propose method.
The results are almost well done.
Now, I present my comments and suggestions as follows:

(1) Why are Poisson mean parameters chosen to take the form of equation

(2) The random effect models should be compared to other methods.

(3) The picture should be modified more clearly. Several figures should be added to the paper to further illustrate the effectiveness of the method

(4) The authors have not fully appreciated numerous interesting references, such as COVID-19 Modelling by fractional calculus (Chaos, Solitons & Fractals, 2020, DOI: 10.1016/j.chaos.2020.110107; ISA Transactions, 2022, DOI: 10.1016/j.isatra.2022.01.008).            

(5) The section discussion should be revised to increase the conclusion of the paper.

(6) In Table 3, fitted death is significantly larger in Australia than in other countries. Is this normal?

Author Response

  1. We have given additional explanation following equation (2)
  2. Thank you for this suggestion. We have considered what we could do here. It remains the case that the most suitable method to compare our methods with are raw (crude) rates. We are now giving a second example, in Section 4.4, to expand on this comparison. We do not feel able to add comparisons with other methods in the present paper. Most alternative methods appear to be based on different data layouts, or make use of implementations which are not available, or provide estimates of "something else".  Adapting any of these methods so that they enable a meaningful and fair comparison would require considerable additional research.  
  3. A second exampe with six additional pictures has been added.
  4. Additional references have been added to the paper which refer to alternative work [23, 24, 31], and have been set it into the context of our methodology.
  5. Thank you for the suggestion. Section 5 has been expanded.
  6. Thank you for this observation. Yes, this is correct and somewhat logical, since the observed deaths for this country (see Table 2) are also much larger than for the other countries.

Round 2

Reviewer 3 Report

I think this paper can be accpeted for publication.